# *MotifDisco*: Motif Causal Discovery For Time Series Motifs

## Abstract

Many time series, particularly health data streams, can be best understood as a sequence of phenomenon or events, which we call *motifs*. A time series motif is a short trace segment which may implicitly capture an underlying phenomenon within the time series. Specifically, we focus on glucose traces collected from continuous glucose monitors (CGMs), which inherently contain motifs representing underlying human behaviors such as eating and exercise. The ability to identify and quantify *causal* relationships amongst motifs can provide a mechanism to better understand and represent these patterns, useful for improving deep learning and generative models and for advanced technology development (e.g., personalized coaching and artificial insulin delivery systems). However, no previous work has developed causal discovery methods for time series motifs. Therefore, in this paper we develop *MotifDisco* (**motif disco**very of causality), a novel causal discovery framework to learn causal relations amongst motifs from time series traces. We formalize a notion of *Motif Causality (MC)*, inspired from Granger Causality and Transfer Entropy, and develop a Graph Neural Network-based framework that learns causality between motifs by solving an unsupervised link prediction problem. We also integrate MC with three model use cases of forecasting, anomaly detection and clustering, to showcase the use of MC as a building block for other downstream tasks. Finally, we evaluate our framework and find that Motif Causality provides a significant performance improvement in all use cases.

## 1 Introduction

Many time series can be best understood as sequences of phenomenon or events. This is extremely common in many health data streams where traces are guided by underlying human physiology or behaviors. We call these events in the traces *motifs*. A time series motif is a short trace segment which may implicitly capture an underlying behavior or phenomenon within the time series. To contextualize our discussion, and as our main running example, we focus on glucose traces collected from continuous glucose monitors (CGMs) for diabetes. Glucose traces inherently contain motifs which represent underlying human behaviors. For instance, a motif capturing a peak in glucose may correspond to an individual eating; a motif capturing a drop in glucose may correspond to an individual exercising. Figure 1 shows some real sample glucose traces and motifs.

Causal discovery is the process by which causal relations are found amongst observational data (Niu et al., 2024). The ability to discover and quantify causality amongst motifs can provide a mechanism to better understand and represent these patterns, useful in a variety of applications. For instance, learning causal relations in glucose motifs may enable better understanding of physiological patterns contributing to advanced technology development (e.g., artificial insulin delivery systems). Moreover, motif causal relationships may be helpful building blocks when used as sub-components in generative and deep learning models to improve their performance for other downstream tasks.

Granger Causality (Granger, 1969) and its nonlinear extension Transfer Entropy (TE) (Schreiber, 2000) are commonly used in time series causal discovery methods, as causal relations are quantified based on one trace's *predictability* of another. This notion of causality provides an intuitive, understandable measure to quantify causal relations and is advantageous for time series because it innately incorporates temporality without requiring strong model assumptions. Despite exciting recent developments for time series causal discovery (Gong et al., 2023; Assaad et al., 2022), no previous

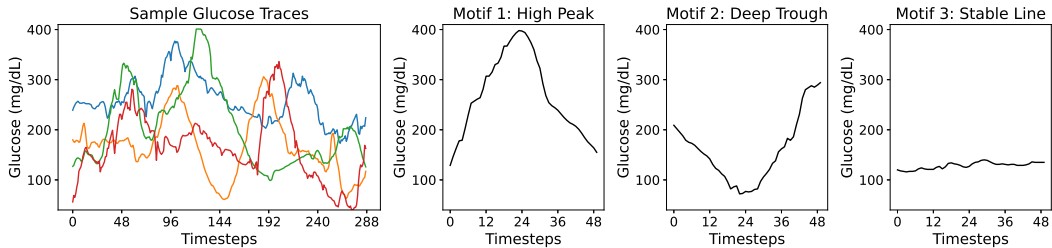

Figure 1: Real Glucose Traces and Sample Motifs for $\tau = 48$.

work has focused on causal discovery for time series *motifs*. Motif causal discovery is challenging because in many cases (particularly in health) there is no ground truth about what underlying behavior a motif captures. For example, from data alone one cannot conclusively determine what caused a change in glucose (e.g., a glucose rise may be from eating *or* stress). As a result, unlike in many other causal models, there is no ground truth causal structure amongst motifs that could be used to guide the casual discovery model (i.e., through supervised methods using labeled causal events.)

Therefore, in this paper we develop ***MotifDisco***, (**motif disco**very of causality), a framework to discover causal relations from time series motifs. First, we formalize the concept of motifs and define a notion of *Motif Causality* (MC) inspired from Granger Causality and Transfer Entropy, which is able to characterize causal relationships amongst sequences of motifs. Next, we develop a causal discovery framework to learn MC amongst a set of ordered motifs pulled from time series traces. The framework uses a Graph Neural Network (GNN) based architecture that learns causality amongst motifs by solving an unsupervised link prediction problem, thereby not requiring knowledge of any ground truth causal structure for training. The framework outputs a directed causal graph where nodes represent motifs and edges represent the degree of the MC relationship. To demonstrate the suitability of Motif Causality as a building block in other models for downstream tasks, we instantiate three model use cases that incorporate MC for forecasting, anomaly detection, and clustering tasks. Finally, we evaluate ***MotifDisco*** in terms of scalability and use case performance by comparing the models with and without integration of MC, to see how helpful MC is for each use case.

The contributions of this paper are: (1) We formalize a new notion of causality between time series motifs, denoted as Motif Causality. (2) We develop ***MotifDisco***, the first causal discovery framework to learn Motif Causality amongst time series motifs. (3) We illustrate the use of MC as a building block in other downstream tasks by integrating MC with three model use cases of forecasting, anomaly detection and clustering. (4) We provide detailed framework evaluation and find that MC provides a significant performance improvement compared to the base models in all use cases.

## 2 RELATED WORK

**Motifs.** Recently, there has been interest in temporal *network motifs*, which are sets of recurring graph substructures. Previous work has investigated network motif causality (Liu et al., 2021; Kovanen et al., 2011) and developed network motif causal discovery frameworks (Chen & Ying, 2024; Chen et al., 2023b; Jin et al., 2022). Importantly, the definition of motif used here is different than ours, referring to patterns in graph structures as opposed to patterns in the traces themselves. In other motif application areas, Chinpattanakarn & Amornbunchornvej (2024) solve a different problem and develop a method to infer a set of patterns, also called motifs, that follow each other in the traces. Finally, Lamp et al. (2024) develop a method to generate synthetic time series glucose traces and use a notion of causality amongst motifs to help the model perform well. The focus of this work is not on causality, and the causal learning method is complicated and suffers from scalability issues.

**Causal Discovery in Time Series.** There are a variety of works on causal discovery for time series (Niu et al., 2024; Gong et al., 2023; Hasan et al., 2023; Assaad et al., 2022; Shojaie & Fox, 2022). In particular, previous methods have developed causal discovery frameworks for multivariate time series that incorporate temporal dynamics and use Granger Causality (Pan et al., 2024; Löwe et al., 2022) or Transfer Entropy (Bonetti et al., 2024; Najafi et al., 2023). Recent work has also

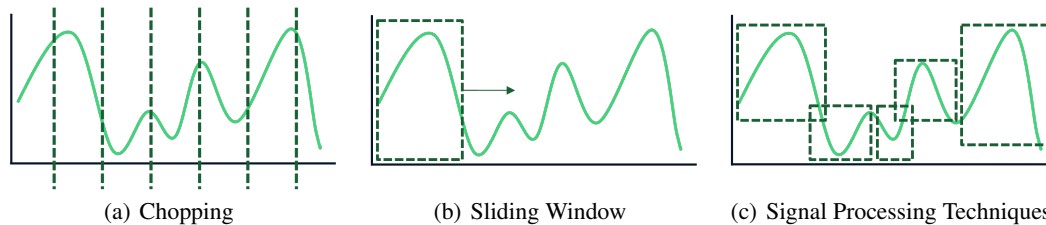

(a) Chopping          (b) Sliding Window          (c) Signal Processing Techniques

Figure 2: Example Motif Construction Methods: (a) chopping the trace into chunks, (b) using a sliding window, or (c) using signal processing techniques to automatically identify motifs.

incorporated time series causality measures to improve model learning in downstream tasks such as forecasting and anomaly detection (Ansari et al., 2024; Chen et al., 2023a; Febrinanto et al., 2023; Duan et al., 2022; Wu et al., 2021). Previous methods for time series causal discovery cannot be directly applied to motifs because they formulate causality using multivariate statistical properties (e.g., variable-based correlation) or temporal statistical measures repeated across time series lags, which do not hold for short, univariate time series motifs that do not contain repeated lagged patterns; or use labels or known underlying causal structures, which are not available for our traces and many similar event-based data streams. ***MotifDisco*** is the first framework for causal discovery in time series motifs, which may broaden current model capabilities for event-based time series.

## 3 FORMALIZING MOTIF CAUSALITY

**Motifs & Motif Construction.** We first formalize our notion of motifs. A motif is a short segment of the trace which may implicitly capture an underlying behavior within a time series. Real sample glucose traces and motifs are shown in Figure 1. We define a *motif*, $\mu$, as a short, ordered sequence of values ($v$) of length $\tau$:

$$\mu = [v_i, v_{i+1}, \ldots, v_{i+\tau}] \tag{1}$$

We denote a set of $n$ time series traces as $X = [x^1, \ldots, x^n]$. Each time series may be represented as a sequence of motifs: $x^j = [\mu_1^i, \mu_2^i, \ldots]$ where each $\mu_t^i$ gives the motif identifier $i$ at the ordered time step $t$. We also define a motif set $\mathcal{M}$, of $|m|$, which is the complete set of motifs generated from the traces $\mathcal{M} = \{\mu^1, \ldots, \mu^m\}$. The user may choose how they wish to pull out motifs from the time series based on the end application goal. We assume there is a consistent, conclusive way to pull motifs from the traces, and motif discovery is outside the scope of this work. We refer the interested reader to other works focused on this problem (Chinpattanakarn & Amornbunchornvej, 2024; Schäfer & Leser, 2022; Ye & Keogh, 2009). That being said, three straightforward methods to extract motifs from traces, shown in Figure 2, include chopping the traces into size $\tau$ chunks (a), using a sliding window of size $\tau$ to extract motifs (b), and using signal processing techniques such as Discrete Fourier Transforms (DFT) to automatically extract motifs (c).

**Granger Causality & Transfer Entropy.** Granger Causality is a common method to characterize causal relations amongst time series (Granger, 1969). Different from other causal methods, Granger defines causal relations in terms of *predictability*. Under Granger Causality, given two time series $x$ and $y$, $x$ causes $y$ if past information about $x$ is more predictive than past information about $y$ only. Transfer Entropy (TE), sometimes also called Causation Entropy, is a nonlinear extension of Granger Causality (Schreiber, 2000; Barnett et al., 2009). Using information theoretic measures, TE measures the amount of uncertainty that is reduced in future states of time series $y$ as a result of knowing the past states of time series $x$. Given two time series, $x^i$ and $y^i$, the TE from $x^i$ to $y^i$ is:

$$TE_{x^i \to y^i} = H(y_t^i | y_{t-1,\ldots,t-f}^i) - H(y_t^i | y_{t-1,\ldots,t-f}^i, x_{t-1,\ldots,t-g}^i) \tag{2}$$

where $H(\cdot|\cdot)$ is a conditional entropy function and $f$ and $g$ are lag constants.

Traditional TE is under the important assumption that the effect is influenced by the cause under a fixed, constant time delay. However, this assumption does not hold for many real world time series applications, particularly in health streams, where data may be affected by past events at

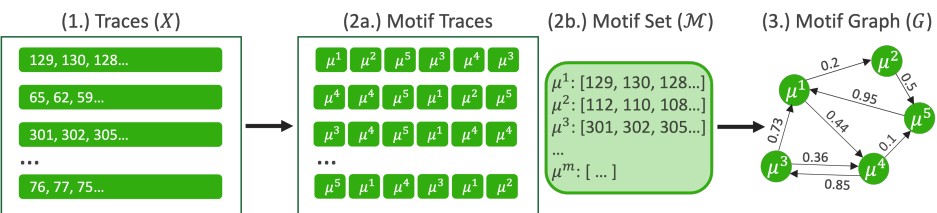

Figure 3: Preprocessing Steps. $X$ are transformed to motif traces and $\mathcal{M}$ and then into a graph $G$.

*varying* lengths of time. As such, an extension of TE that allows for different time delays has been developed, denoted here as Variable-lag Transfer Entropy (VTE) (Amornbunchornvej et al., 2021). The VTE from $x^i$ to $y^i$ is defined as:

$$VTE_{x^i \to y^i} = H(y_t^i | y_{t-1,...,t-f}^i) - H(y_t^i | y_{t-1,...,t-f}^i, x_{t-1-\Delta_{t-1},...,t-g-\Delta_{t-g}}^i) \tag{3}$$

where $\Delta_t$ is a variable length lag amount. We will adapt this equation and other notions of TE next for our definition of Motif Causality.

**Defining Motif Causality.** Our definition of motif causality is inspired from various Transfer Entropy and Causation Entropy threads (Equation 3, Irribarra et al. (2024); Gong et al. (2023); Amornbunchornvej et al. (2021); Assaad et al. (2021); Sun et al. (2015)). The Motif Causality (MC) from motif $\mu^i$ to motif $\mu^j$ conditioned on the set of motifs $\mathcal{K}$ is defined as:

$$MC_{\mu^i \to \mu^j | \mathcal{K}} = H(\mu_t^j | \mathcal{K}_{t-1,...,t-f}) - H(\mu_t^j | \mathcal{K}_{t-1,...,t-f}, \mu_{t-1-\Delta_{t-1},...,t-g-\Delta_{t-g}}^i) \tag{4}$$

where $\mathcal{K} \subset \mathcal{M}$, $H(\cdot | \cdot)$ is a conditional entropy function, $f$ and $g$ are lag constants and $\Delta_t$ is a variable length lag amount. Essentially, this provides a measure of information gain by determining how much uncertainty is reduced for $\mu^j$ by observing past occurrences of $\mu^i$ compared to the "status quo", the set of the rest of the motifs $\mathcal{K}$. The $MC$ value will be between 0.0 and 1.0. Higher values indicate stronger causality (i.e., more uncertainty about the future is reduced for $\mu^j$ given $\mu^i$).

**Conditional Entropy Function.** To implement the conditional entropy function, $H(\cdot | \cdot)$, there are many different types of entropy which the user can choose based on their end goal or application. For our purposes, we elucidate two common ones: Shannon entropy (Shannon, 1948) and Rényi entropy (Jizba et al., 2012; 2022). Shannon entropy is defined as:

$$H(x^i) = -\sum_t p(x_t^i) \log_2(p(x_t^i)) \tag{5}$$

where $p$ is a probability distribution. Rényi is more flexible at estimating uncertainty and defined as:

$$H_\alpha(x^i) = \frac{1}{1-\alpha} \log \left( \sum_{v=1}^n p_v^\alpha(x^i) \right) \tag{6}$$

where $\alpha$ is a weight parameter, $\alpha > 0$. When $\alpha \to 1$ Rényi entropy converges to Shannon entropy. We note that there are many other types of entropy functions that could be used and might be relevant, such as Wavelet (Rosso et al., 2001) or Permutation Entropy (Bandt & Pompe, 2002).

## 4 MOTIF CAUSAL DISCOVERY FRAMEWORK

Now that we have formulated our definition of Motif Causality, we next describe our casual discovery framework ***MotifDisco*** to learn motif causal relationships amongst a set of time series motifs. We first detail preliminaries related to the problem definition and preprocessing in Section 4.1, describe the model architecture in Section 4.2 and finish with the model training algorithm in Section 4.3.

### 4.1 PRELIMINARIES

**Problem Definition.** Since we do not know nor have any way to determine the underlying motif-causal structure (i.e., we have no ground truth), we formulate this problem as an unsupervised graph

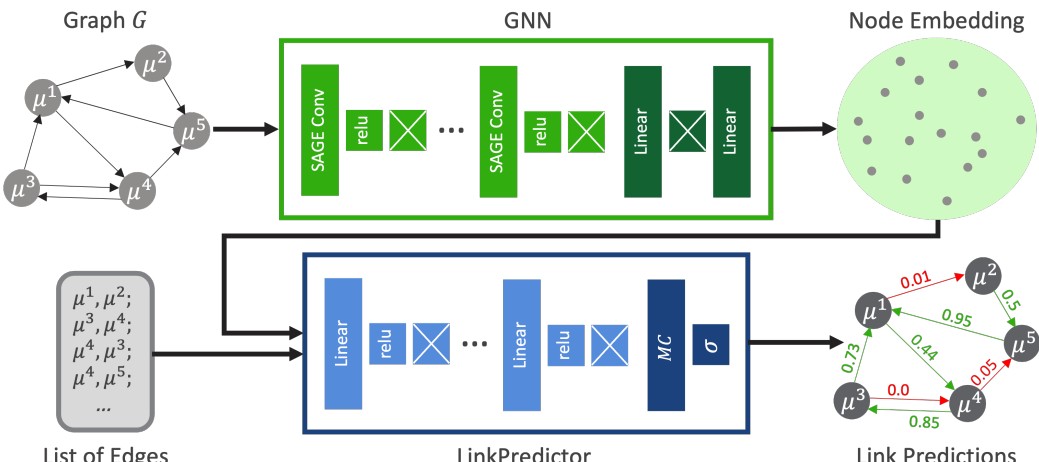

Figure 4: Model Architecture: a GNN consisting of stacks of GraphSAGE, ReLU, Dropout ("X" boxes) and Linear layers learns a node embedding. The LinkPredictor consists of Linear, ReLU and Dropout layers followed by the "MC" motif causality layer and a Sigmoid layer ($\sigma$ box). The LinkPredictor takes in the node embedding and outputs the predicted edges and their weights.

link prediction problem: Given a set of input time series motifs extracted from our set of traces $X$, and the complete set of nodes which is equivalent to the motif set $\mathcal{M}$ (i.e., each motif is a node), predict the edges between all the nodes. In other words, learn the edge weights, the motif causal relationships, between all the nodes, the motifs. To solve this problem, we build a Graph Neural Network-based causal discovery framework that learns the MC edge weights in an unsupervised manner. We walk through each part of the framework next, starting with the preprocessing steps.

**Preprocessing.** An overview is shown in Figure 3. Motifs of length $\tau$ are pulled from the input time series to create a set or ordered motif traces and the motif set $\mathcal{M}$ (see Section 3). In our implementation, we use the chopping method to generate motifs; $\mathcal{M}$ is the union of all size-$\tau$ chunks in the traces. From there, an initial motif graph is generated from the motif traces. In the graph structure, each node contains the motif identifier and the actual motif values (e.g., motif $\mu^1 = [129, 130, 128, ...]$). Edges represent directed motif-causal relationships between two motifs. The edge weight indicates the *strength* of the causal relationship. At this stage, generating the *best* graph is not our primary concern since the framework will add and remove optimal edges during the learning process, and we just need a starting graph structure to build from. As such, we believe it an acceptable first pass to assume there is *some* degree of causality between motifs that appear immediately one after the other, and build the preliminary graph by adding edges between each subsequent motif in the traces. For example, for the first motif trace in Fig. 3 (2a), edges would be added from $\mu^1 \rightarrow \mu^2$, $\mu^2 \rightarrow \mu^5$, etc. We compute the MC edge weight according to Equation 4.

## 4.2 MODEL ARCHITECTURE

An overview of the model architecture is shown in Figure 4. The model consists of two main components: a Graph Neural Network (GNN) that uses GraphSAGE layers to learn node embeddings, and a LinkPredictor network that uses the learned node embeddings to predict motif causal links amongst nodes. We detail each component next and then describe model training in Section 4.2.

**GNN.** The Graph Neural Network takes in a starting motif graph $G$, and outputs a learned node embedding. The GNN is structured using GraphSage convolutional layers. GraphSAGE (standing for Sample and Aggregate) (Hamilton et al., 2017) learns low dimensional vector representations of nodes. The core intuition of GraphSAGE is that a node is known by the company it keeps (i.e., its neighbors). The algorithm works by iterating over a sample of the node's neighboring nodes and "aggregating" their embeddings in order to determine the current node's embedding. Importantly, GraphSAGE is *inductive*, meaning it can generalize to unseen nodes. GraphSage use both node

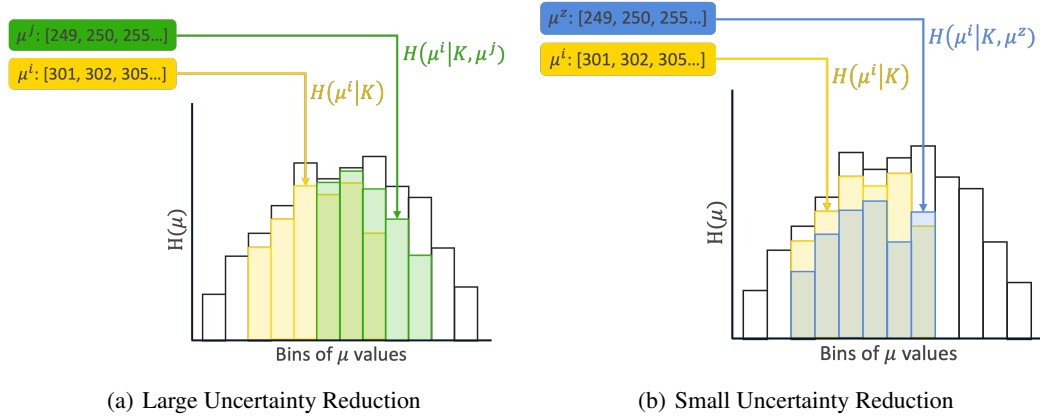

(a) Large Uncertainty Reduction         (b) Small Uncertainty Reduction

Figure 5: Depiction of Conditional Entropy $H(\mu)$ computation using the histogram method: when computing MC for $\mu^i$, (a) $\mu^j$ results in a large uncertainty reduction; (b) $\mu^z$ has a small reduction.

features and topological structure (i.e., the graph structure) of each node's neighbourhood simultaneously to efficiently generate representations for new nodes without requiring model retraining. To do this, the algorithm relies on its aggregation function, also known as the message-passing process, which learns how to aggregate node features based on encoding information about a node's local neighborhood. Therefore, when given new node data, the function uses the local neighborhood of the node to aggregate the features appropriately, and learn the embedded feature representation (as opposed to needing to learn a unique embedding for every individual node). As shown in Figure 4, in our framework the GNN consists of sequential stacks of GraphSAGE convolutional, ReLU and dropout layers (represented by the "X" boxes in the figure) followed by the message-passing layers–stacks of linear and dropout layers. The motif time series values are the node data used to learn the embeddings. We instantiate the aggregation function using mean aggregation for simplicity.

**Link Predictor.** The LinkPredictor network takes in learned node embeddings outputted from the GNN and a list of edges to predict, and returns the probability of each edge and the motif causality values for each edge. At its core, the LinkPredictor learns a function to predict the probability of an edge between two nodes. To do this, it computes the MC between the nodes using Equation 4, and a probability score, represented by the element-wise dot product of the two embedded node vectors. In terms of architecture, the LinkPredictor is implemented via stacks of linear, ReLU and dropout layers followed by Motif Causality ($MC$) and sigmoid layers to learn the edge probability function. The network balances evaluating the product of the embedded node vectors and the motif causal values between the motifs themselves, allowing the network to learn how to optimize edge addition/deletion in the graph guided by the underlying causality. (More details about this in Section 4.3). In this way, the LinkPredictor learns to predict edges that have high motif causality with high probability.

To implement the MC computation, we adapt existing Transfer Entropy libraries (Behrendt et al., 2019) and compute the conditional entropy function (i.e., $H(\mu)$, Shannon's or Rényi entropy) using the histogram method. A simplified depiction is shown in Figure 5. Essentially, motif time series values are binned into a histogram. Distributions between motifs can be compared to determine how much uncertainty about future predictions of the motif is reduced in the distribution. For example, in 5(a) when computing MC for $\mu^i$, $\mu^j$ covers a larger distribution, resulting in a large reduction in uncertainty and higher motif causality, whereas 5(b) $\mu^z$ covers hardly any new distribution compared to the status quo ($H(\mu^i|\mathcal{K})$), resulting in a small reduction in uncertainty and low causality for $\mu^z$.

### 4.3 MODEL TRAINING

We next describe how the entire framework is trained together, shown in Algorithm 1. First, the graph is fed through the GNN network to learn an embedded representation of the nodes (Line 2). Next, the LinkPredictor makes predictions on a sample of the edges that exist in the graph $G$, denoted as the positive edges $E$ using the learned node embedding (Line 3-4). Then, a batch of edges that

---

**Algorithm 1:** Training Procedure to Learn Motif Causality

---

**Input:** Input Graph $G$, Epochs $e$, Edge Prediction Threshold $\theta$

**1 for** *e epochs* **do**

    /* Compute Node Embedding */

**2**     node_emb = **GNN** $(G)$

    /* Get predictions on positive edges */

**3**     $E \longleftarrow sample\,(G.edges)$// Edges that exist in $G$

**4**     $p, c =$ **LinkPredictor** (node_emb, $E$)

    /* Get predictions on negative edges */

**5**     $\hat{E} \longleftarrow negative\_sample\_edges\,(G)$// Edges not in $G$

**6**     $\hat{p}, \hat{c} =$ **LinkPredictor** (node_emb, $\hat{E}$)

    /* Compute Loss */

**7**     $y = \gamma \times p + \lambda \times c$

**8**     $\hat{y} = 1 - (\gamma \times \hat{p} + \lambda \times \hat{c})$

**9**     $loss = -\sum_{i=1}^{b} (y_i \log \hat{y}_i + (1 - y_i) \log(1 - \hat{y}_i))$

    /* Update edges in the graph based on new predictions */

**10**     $G.update\_edges\,(E, y, \hat{E}, \hat{y}, \theta)$

**11 end**

---

do not exist in $G$, $\hat{E}$, are randomly sampled by selecting random pairs of nodes with no connections between them (Line 5). The LinkPredictor makes predictions on the negative edges (Line 6).

From there, the positive predictions $y$ are computed by combining the positive edge predictions $p$ and the motif causality values for the positive links $c$ (Line 7). Similarly, the negative predictions $\hat{y}$ are computed by combining the negative link predictions $\hat{p}$ with the MC values for the negative links $\hat{c}$ (Line 8). $\gamma$ and $\lambda$ are important hyper-parameters which balance the influence of the dot product predictions vs. the motif causality. The model is trained to minimize the log likelihood loss function, computed following the equation on Line 9. Essentially, this loss function optimizes the model to maximize its predictions of positive edges (true links, edges that have high motif causality values) and minimize predictions of negative edges (links that should not exist in $G$, edges with low MC). Finally, the edges in $G$ are updated based on the model edge predictions and an edge threshold $\theta$ (Line 10). $\theta$ is a user-specified parameter with a range between 0 and 1. If any of the edges in $\hat{y} \geq \theta$, they are added to the graph $G$; if any of the edges in $y < \theta$, they are removed from $G$.

Due to the dual training between the GNN and the LinkPredictor, as the model iterates the node embeddings are continuously updated based on new linkages that may be added or removed. As such, the GNN learns good node embeddings representative of the types of nodes that are close to each other and have connections to each other (nodes that are very motif causal of one another). The LinkPredictor is guided by the motif causality values between motifs to help it predict where linkages should be, and as it learns characteristics of highly causal edges, and gets better and better node embeddings from the GNN, it learns a good prediction function for predicting node linkages.

**Making New Edge Predictions.** To make a link prediction between two new nodes using the trained framework, the nodes are first fed through the GNN to get their node embeddings, and then the embeddings are sent through the trained LinkPredictor network, which returns the predicted probability they have an edge between them, along with the motif causality value.

## 5 USES OF LEARNED MOTIF CAUSALITY

To illustrate the use of Motif Causality as a building block for other downstream tasks, in this section we integrate Motif Causality with three model use cases of Forecasting, Anomaly Detection and Clustering. We evaluate the performance of these use cases later in Section 6.

**Forecasting.** Our integration is shown in Figure 6. Briefly, some basic forecasting models work by sliding a window across input traces to learn sequential time series patterns. Their loss function computes the difference between the model's predicted future timesteps (y_pred) and the ground

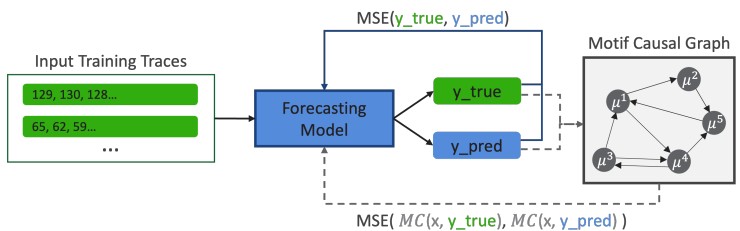

Figure 6: Forecasting Model integrated with Motif Causality.

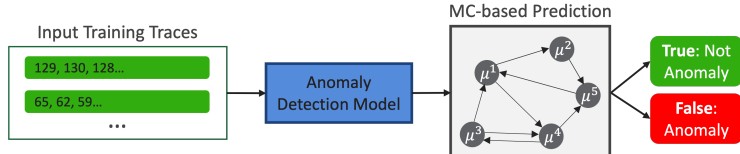

Figure 7: Anomaly Detection Model integrated with Motif Causality.

truth future time steps (y_true) in the traces. To integrate motif causality, we add an additional loss function that seeks to minimize the difference in the MC between the previous time step ($x$) and the predicted future time steps (y_pred) vs. $x$ and the ground truth future time steps (y_true). The intuition is that there should be similar causality between the previous time step and the predicted future time step as the ground truth data. MC is computed using the trained motif causal graph outputted from the MC framework. We assume the motif size $\tau$ is the same as the forecasted prediction window size to ensure MC is computed on comparable time chunks (i.e., motifs).

**Anomaly Detection.** Integration of a basic anomaly detection model with MC is shown in Figure 7. We add an MC-based anomaly prediction block after the model that uses the predicted MC between previous timesteps and future timesteps to determine if the next trace chunk may be anomalous or not. Specifically, it checks if the MC between the previous time steps (previous motif) and the next one are less than a threshold, and if so classifies it as an anomaly. The size of the predicted time chunk must be the same size as the motif size $\tau$ and we suggest the anomaly threshold be set to the edge prediction threshold $\theta$, since this was what was used to train the motif causal graph originally.

**Clustering.** For clustering, there is typically a method to group clusters based on minimizing distances between each data point and the cluster centroid. These distances are computed using various distance measures such as Discrete Time Warping (DTW) (Sakoe & Chiba, 1978) for traces. To integrate MC with a basic clustering algorithm, we use the motif causality values as an additional distance metric. The intuition here is to add an element of *causality* to the clustering, such that as the algorithm learns, MC values within each cluster will be minimized and similar data points within the cluster should be *motif causal* of each other. An example is shown in Figure 11 in A.1: the MC value between the blue centroid and the blue data point to the right is high with 0.9, whereas the MC between the blue centroid and the green data point belonging to a different cluster is low at 0.11.

## 6   EVALUATION

In this section we evaluate our causal discovery framework in terms of scalability and performance for three downstream task use cases: forecasting, anomaly detection and clustering.

**Experimental Details.** For all experiments, we use sets of single-day glucose traces randomly sampled across each month from January to December 2022, collected from Dexcom's G6 Continuous Glucose Monitors (CGMs) Akturk et al. (2021). Data was recorded every 5 minutes (total of 288 timepoints per trace) and each trace was aligned temporally from 00:00 to 23:59. All experiments were run 5 times with an 80/20% train/test split on an Intel Sky Lake 48 CPU VM with 192GB of RAM. Motifs were pulled from traces using the chopping method and we set $\gamma = 0.7$ and $\lambda = 0.5$.

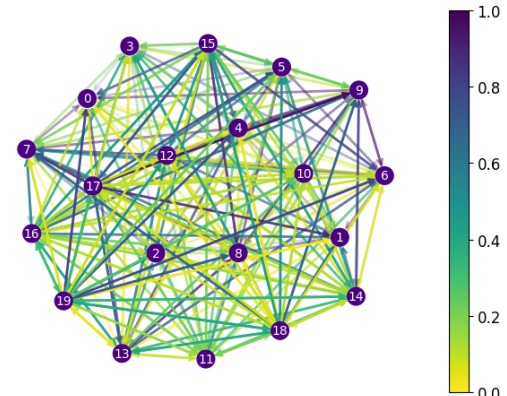

Figure 8: Learned Motif Causal Graph for $n = 10$, $|\mathcal{M}| = 20$. Edge color indicates the MC value.

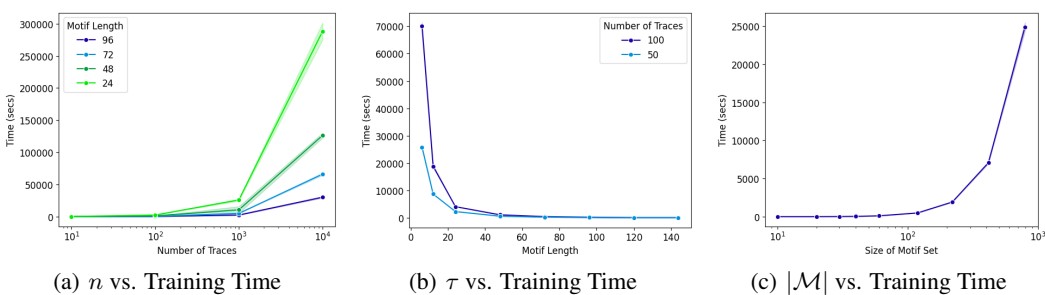

(a) $n$ vs. Training Time      (b) $\tau$ vs. Training Time      (c) $|\mathcal{M}|$ vs. Training Time

Figure 9: Scalability varying (a) # of traces ($n$), (b) motif lengths ($\tau$) and (c) motif set sizes ($|\mathcal{M}|$).

**Motif Causal Graphs.** An example motif causal graph from the ***MotifDisco*** framework is shown in Figure 8. The edge color indicates the strength of the MC relationship; darker is stronger (closer to 1) while lighter is less strong (closer to 0). A.1 shows example MC between motifs of different $\tau$.

**Scalability.** The scalability evaluation is shown in Figure 9, computing total time to train the ***MotifDisco*** causal discovery framework in seconds for 9(a) different numbers of traces ($n$), 9(b) motif lengths ($\tau$) and 9(c) motif set sizes ($|\mathcal{M}|$). In all experiments training was for 10 epochs. As to be expected, training time increases for larger $n$ and $|\mathcal{M}|$. Interestingly, time reduces as the $\tau$ increases, which may in part be because larger $\tau$s mean there are less total motifs (i.e., $|\mathcal{M}|$ is smaller).

**Use Cases.** We next evaluate the suitability of Motif Causality to help in three downstream tasks: Forecasting, Anomaly Detection and Clustering. For each use case, we build a simple base model and integrate MC following the architecture descriptions in Section 5. We then compute a set of evaluation metrics to compare the performance between the base model and the MC integrated one.

**Forecasting.** For the forecasting task, we use a simple bidirectional LSTM as our base model. We set the sliding window size, $\tau$ and forecasting window to 6 (corresponding to 30 minutes of time). We train the causal discovery and forecasting models for 20 and 2000 epochs, respectively. The Root Mean Square Error (RMSE) is reported in Table 1; the MC model outperforms the base one.

**Anomaly Detection.** For this task, we build a simple autoencoder consisting of stacks of sequential dense layers. In the base model, an anomaly is detected if the Mean Absolute Error (MAE) of the reconstructed data (i.e., the trace fed through the encoder and then returned via the decoder) is less than a reconstruction threshold which we set as the standard deviation of the mean of the normal (non-anomalous) training data. In the MC-integrated version, we detect an anomaly if the predicted MC value between the previous time chunk (motif) and the current one is less than the edge prediction threshold $\theta$. We set the sliding window size and $\tau$ to 48, $\theta$ to 0.1 and train the causal discovery and anomaly models for 10 and 50 epochs, respectively. We compare the models by computing

Table 1: Use Case Performance Summary. The arrow indicates desired result direction and bold values indicate the best performing model.

| Model | Forecasting RMSE (↓) | Anomaly Detection Accuracy (↑) | F1 (↑) | Sensitivity (↑) | Specificity (↑) | Clustering C-Index (↓) | SSE (↓) | Silhouette (↑) | Caliński Harabasz (↑) |
|---|---|---|---|---|---|---|---|---|---|
| Base | $0.22 \pm 0.05$ | $0.83 \pm 0.02$ | $0.80 \pm 0.03$ | $0.67 \pm 0.02$ | $\mathbf{1.0 \pm 0.0}$ | $0.086 \pm 1e{-}6$ | $477.0 \pm 1e{-}3$ | $0.318 \pm 1e{-}2$ | $490.95 \pm 6e{-}2$ |
| MC | $\mathbf{0.18 \pm 0.07}$ | $\mathbf{0.92 \pm 0.02}$ | $\mathbf{0.92 \pm 0.02}$ | $\mathbf{1.0 \pm 0.0}$ | $0.85 \pm 0.03$ | $\mathbf{0.085 \pm 3e{-}5}$ | $\mathbf{473.79 \pm 5e{-}3}$ | $\mathbf{0.332 \pm 2e{-}3}$ | $\mathbf{496.24 \pm 5e{-}4}$ |

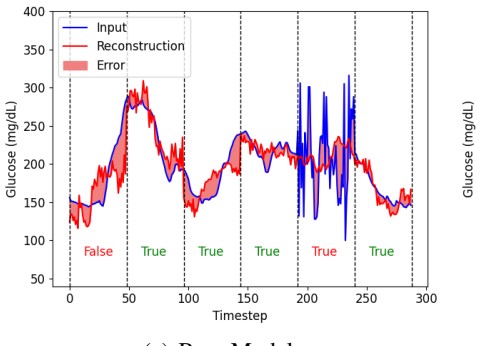
(a) Base Model

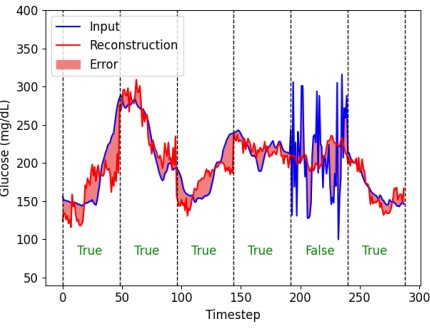
(b) Base Model + MC

Figure 10: Anomaly Detection for a trace with an obvious anomaly using the (a) base model and the model integrated with MC (b). False means an anomaly was predicted and the color corresponds to correctness of the prediction: green is a correct prediction, red is an incorrect one.

a set of classification metrics including accuracy, F1, Sensitivity and Specificity using the ground truth labeled anomalies. Results are reported in Table 1. For all metrics except Specificity, the MC-integrated model does better. Interestingly, the base model has better Specificity while the MC model has better Sensitivity - this indicates the MC model is better at identifying the anomalies (the true negatives) and the base is better at identifying the normal traces (the true positives). Example anomaly predictions are shown in Figure 10. Each graph plots the original input trace (in blue) and the reconstructed trace (in red, trace fed through the encoder + decoder) with the error between the two shaded in red. The window segments (dashed black lines) correspond to the motif and prediction window size and the model predictions are annotated in each window, colored by the correctness of the prediction. Green indicates a correct prediction, red indicates an incorrect one.

**Clustering.** We implement a simple K-Means clustering algorithm. In the base model we compute distances between the cluster centroids and other trace data points using Discrete Time Warping (DTW) (Sakoe & Chiba, 1978). In the MC-integrated version we compute the distances as DTW + MC. We set the number of clusters $k = 3$, and $\tau = 48$. Causal discovery and clustering models were trained for 20 epochs. We compute a set of clustering evaluation metrics including C-Index (Hubert & Levin, 1976), Sum of Squared Error (SSE) (Macqueen, 1967), Silhouette score (Rousseeuw, 1987), and Caliński Harabasz score (Caliński & Harabasz, 1974), reported in Table 1. Across all metrics the MC version does better, indicating adding an element of causality may help clustering.

# 7 CONCLUSION & LIMITATIONS

In this paper we presented *MotifDisco*, the first causal discovery framework to infer Motif Causality amongst time series motifs. By providing a new method to learn and quantify relationships amongst motifs, *MotifDisco* may facilitate the development of advanced, high performing technologies for event-based time series. As shown by the scalability experiments, for very large $\mathcal{M}$ and $n$ (e.g., $n \geq 10000$) the runtime can take several hours. There are many opportunities in the training framework to further optimize the runtime. For example, MC is currently computed between each edge in a batch sequentially; using sampling or parallelization would significantly speed up the training time. Additionally, a challenge of this work is that there is no known causal structure available, so it was not possible to evaluate the learned motif causal graphs against some ground truth. However, as evidenced by the use case evaluation, using a relatively simple integration of MC with naive base models resulted in significant performance improvements for all three use cases, providing some evidence about the potential generalizability and applicability of MC for many real world tasks.

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

# A  APPENDIX

## A.1  ADDITIONAL FIGURES

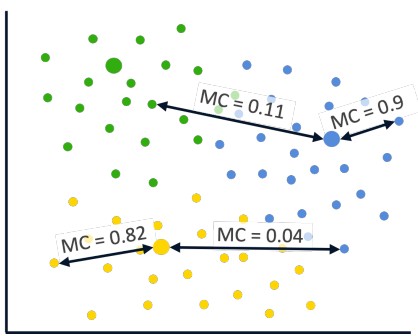

Figure 11: Depiction of Clustering Model integrated with MC.

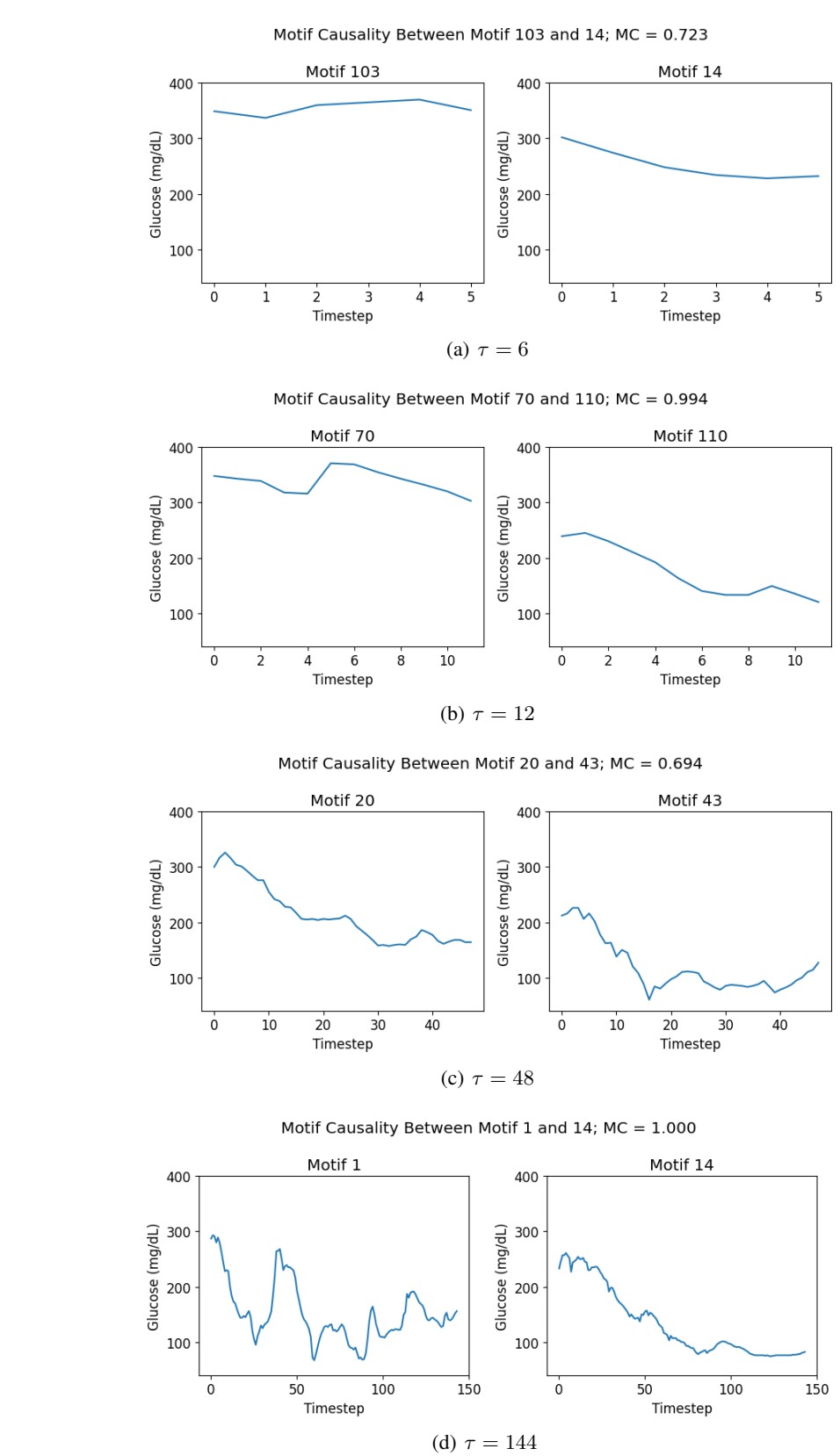

Figure 12: Example High Motif Causality values between different motif sizes.

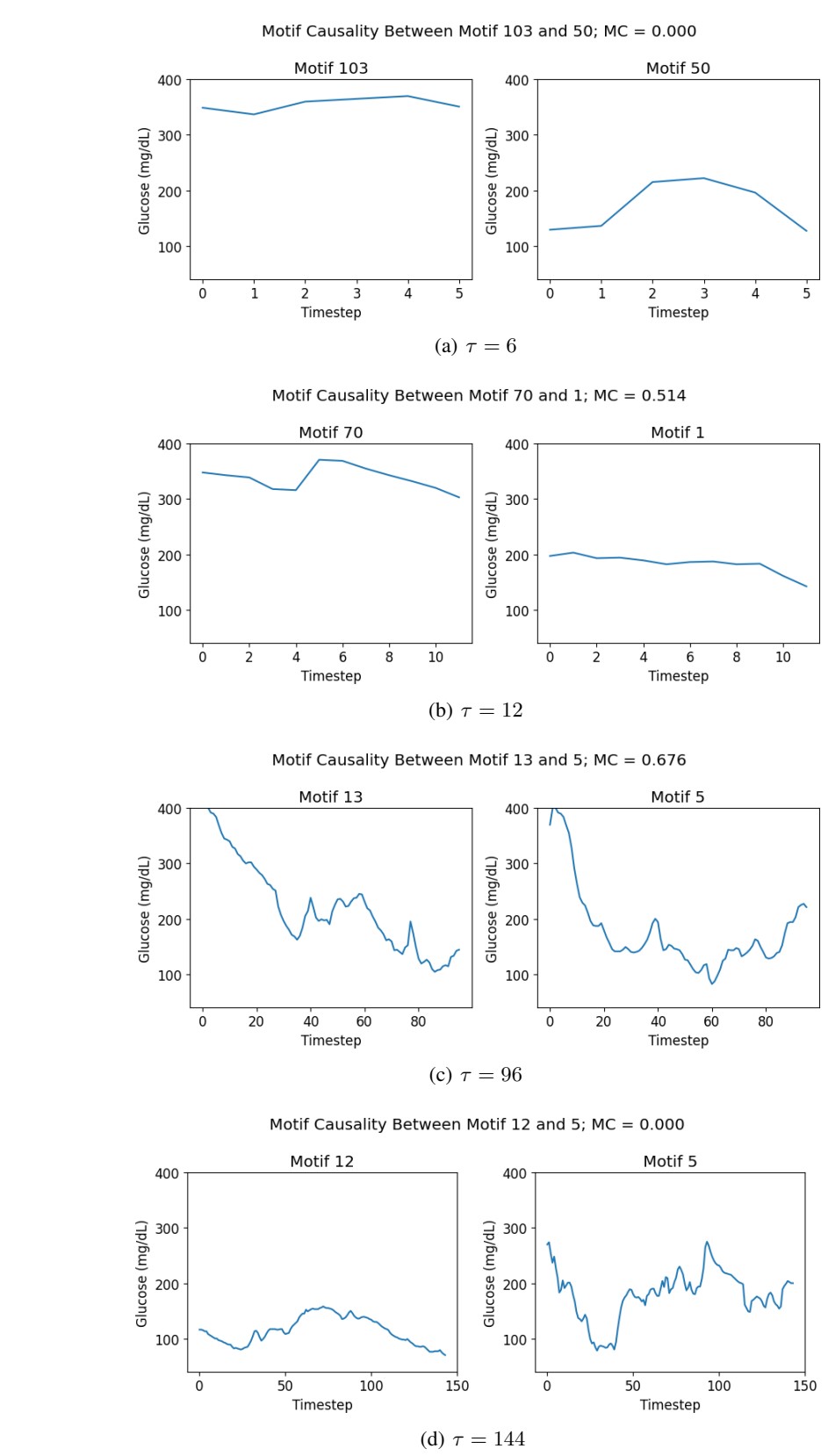

Figure 13: Example Low Motif Causality values between different motif sizes.

