# OpenReview forum: "MotifDisco: Motif Causal Discovery For Time Series Motifs"
_ICLR.cc/2025/Conference — Submitted to ICLR 2025_

### Official Review · Reviewer_FkcT · 2024-10-30

**Soundness:** 3
**Presentation:** 2
**Contribution:** 2
**Rating:** 5
**Confidence:** 3

**Summary:**

In this paper author proposed MotifDisco (motif discovery of causality), a novel causal discovery framework to learn causal relations amongst motifs from time series traces based on Granger Causality and Transfer Entropy. Used motif causality in down stream tasks like forecasting, anomaly detection and clustering.

**Strengths:**

The paper is well written. The literature survey is good. Work is mathematically sound and the author shows run time requirements. The idea of showing the model performance on three tasks was also good.

**Weaknesses:**

Marginal technical novelty. What is the contribution compared to  Lamp et el (2024) needs to be discussed. Pan et al., 2024; Lowe ¨ et al., 2022 Bonetti et al., 2024; Najafi et al., 2023) are already using  Granger Causality and Transfer Entropy then what is contribution compared to this work is not clear to me.
Please explicitly state key technical contributions of this paper and how it differ from or improve upon the cited works, particularly in the context of motif-based causal discovery for time series.

I could not understand what you mean by discovery. I am requesting a clear definition of what the authors mean by "discovery" in this context would help clarify the paper's novelty.

The experiment section is incomplete. The author has compared with only one base model.  Also base model is built by the author. The author needs to provide the architecture of the base model. The author need to compare the work with other existing state of art motif causality models for all tasks like Pan et al., 2024; Lowe ¨ et al., 2022 Bonetti et al., 2024; Najafi et al., 2023.
Also, it seems the author has compared the proposed model only for one data set for each task. Please compare on more real-world data sets. Also, provide details of the data set used.

**Questions:**

As above

---

> ### Author Response · Authors · 2024-11-13
> **Response to Reviewer FkcT**
>
> Thanks for your time and effort on this review.
>
> Technical Novelty:
> MotifDisco is the first framework to learn causal relationships amongst time series motifs, and there is no prior work solving this problem. In this context, Motif “discovery” of casuality refers to the process of identifying previously unknown causal relationships amongst a set of motifs. Lamp et al, 2024 solve a different problem (they are trying to generate synthetic data), quantify causality differently (i.e., using a different formulation) and the process they use to learn causality is complex, not interpretable and has issues with scalability. In addition, previous methods for causality in time series cannot be directly applied to learn causal relations amongst motifs because of one or more of the following issues:
> -	They compose relations between only two time series, and cannot find relationships amongst a set of motifs or traces (Amornbunchornvej et al., 2021; Gong et al., 2023).
> -	They formulate causality based on various statistical properties amongst variables- things like variable-based correlation and density-, classification- and prediction-based error measures between variables (Amornbunchornvej et al., 2021; Bonetti et al., 2024; Irribarra et al., 2024). These assumptions do not hold for univariate time series motifs.
> -	They compose causal relationships over time series using repeated statistical measures or temporal dynamics – things like learning implicit repeated patterns, or computing sufficiency or faithfulness hypotheses over lags across the full time series (Sun et al., 2015; Assaad et al., 2021; Löwe et al., 2022; Najafi et al., 2023; Lamp et al., 2024; Pan et al., 2024). These methods do not work for motifs because they are short time series (so there would be no repeated patterns, nor will statistical hypotheses like sufficiency or faithfulness hold since they are not evaluating multiple time lags repeated over a trace.).
> -	They require labels or a known underlying causal structure to guide model training (Gong et al., 2023; Najafi et al., 2023; Bonetti et al., 2024), which is not available for our data and many similar event-based data streams.
>
> To make this more clear, we have updated the Related work section to explicitly state this (see lines 122-128).
>
>
> Experimental Evaluation:
> While we do agree that comparing with existing state of the art would make the paper stronger, unfortunately this is not possible because there is no prior work on quantifying causality amongst motifs. We cannot directly compare with prior Granger causal techniques because their framework set-up or model assumptions do not hold for univariate motifs (see more details in the related question previously about technical novelty).
>
> We focused our evaluation to one domain (diabetes, using glucose traces) because this application is particularly well motivated to use and understand relationships amongst motifs. We selected this dataset because it contains a large set of glucose traces with high variability and motif representation amongst traces. Future work may investigate uses of MotifDisco on other health datasets.
>
> Details about the base models used for each use case are provided in Section 5. Details about the dataset used are provided in Section 6 under Experimental Details. If there are additional questions related to use case model architectures or the dataset we are happy to answer them!

---

> > ### Comment · Reviewer_FkcT · 2024-11-13
> > **Thanks**
> >
> > I thank the author for addressing my concern and put additional information in related world section. but I still think experiment with only one real data is not enough.   I adjusted my score accordingly.

---

### Official Review · Reviewer_iup6 · 2024-11-03

**Soundness:** 2
**Presentation:** 3
**Contribution:** 2
**Rating:** 3
**Confidence:** 4

**Summary:**

The paper introduces MotifDisco, a novel framework for motif causal discovery in time series. The authors focus on causal relations among motifs, defined as short segments representing underlying phenomena within time series. MotifDisco leverages a combination of Granger Causality and Transfer Entropy to define Motif Causality (MC) and uses a GNN to learn causal relationships by solving an unsupervised link prediction problem. The framework is evaluated on glucose traces collected from continuous glucose monitors and further integrated into forecasting, anomaly detection, and clustering tasks. Overall, the authors claim significant improvements in performance for each of these downstream applications compared to non-causal baselines.

**Strengths:**

1. Novelty of Causal Discovery Framework: The introduction of Motif Causality (MC) for time series motifs and the development of MotifDisco fills an important gap in time series analysis, especially for health-related data. No prior work has explicitly targeted causal discovery among motifs within time series, which makes this a novel contribution.
2. Flexible Application Scope: Integrating MC into multiple use cases, namely forecasting, anomaly detection, and clustering, demonstrates the proposed framework's versatility and broadens its potential real-world applicability.
3. The experiments span various scenarios, including different motif extraction methods, motif lengths, and scalability. The performance gains in forecasting and anomaly detection validate the utility of incorporating causality into motif-based models.

**Weaknesses:**

1. Motif Construction Limitation: The method for constructing motifs is largely dependent on heuristic techniques (e.g., chopping or sliding windows). This may lead to arbitrary definitions of motifs that do not always correspond to well-defined physiological phenomena. The authors could consider more dynamic motif extraction methods.
2. Lack of Personalization: The majority strategy for causal inference used in the GNN might overlook personalized differences across individuals, which could limit the accuracy of BP estimation or understanding of other health parameters in highly diverse populations.
3. No Ground Truth for Causal Evaluation: A notable limitation is the lack of ground truth causal structures for motifs, which makes the evaluation of the learned causal graphs challenging. Although indirect measures such as downstream task performance are used to validate the usefulness of the model, a more direct assessment of the accuracy of causal inference is missing.\
4. No comparison with SOTA: this paper does not provide an extensive, systematic comparison against other state-of-the-art causal discovery frameworks and deep learning methods for forecasting, anomaly detection, and clustering.
5. Scalability Issues: The scalability analysis shows that training times grow significantly for large motif sets and numbers of traces. The current implementation may not be suitable for very large datasets, especially in real-time applications. Methods like parallelization for computing motif causality are suggested as improvements.
6. Limited Clinical Validation: The evaluation was limited to glucose data, and the clinical significance of the discovered causal motifs is not thoroughly validated. This limits the generalizability of the proposed method to other medical domains without further empirical evidence.

**Questions:**

1. How does the chosen motif extraction method impact the causality results? Would alternative approaches, such as clustering-based motif identification, lead to different outcomes?
2. How does the interpretability of the causal graphs change across datasets with different numbers of motifs and motif lengths?
3. How generalizable is MotifDisco to other health data, such as heart rate variability or electroencephalogram (EEG) signals?
4. How does the framework account for inter-individual variability, especially given that causal relationships can be highly individualized?

---

> ### Author Response · Authors · 2024-11-13
> **Response to Reviewer iup6**
>
> Thanks for your time and detailed review.
>
> Motif Construction (Weakness 1 and Question 1):
> Since we wanted the focus of this paper to be on the formulation and development of the motif causality framework, motif extraction was outside the scope of the paper. That being said, as mentioned by the reviewer (and as also mentioned in our paper), more dynamic or intelligent motif extraction methods could definitely be used for motif extraction based on the desired use case (e.g., methods like Chinpattanakarn & Amornbunchornvej, 2024; Scha ̈fer & Leser, 2022; Ye & Keogh, 2009). Our framework is amenable to any motif extraction method.
>
> Since our framework finds relationships amongst predefined motifs, the results will look different if a different set of input motifs are given. We note that interesting causal relationships were found amongst motifs even when a simple motif extraction method was used, such as chopping.
>
> Graph Interpretability (Question 2):
> Interesting question. It depends on the dataset, but for instance for the glucose traces the causal graph structure will get larger and more complex (e.g., because there may be more edges between more motifs) as the number of motifs increases, and the length of the motifs decreases. A way to balance this is through the edge prediction threshold, \theta. For example, if you wanted a sparser graph that only contained the strongest causal relationships, you could set \theta to a high value such as 0.8.
>
> Generalizability (Question 3):
> MotifDisco is very generalizable, because the framework does not make any glucose-specific assumptions in its preprocessing, learning or post-processing. The only constraint required by the framework is that you provide an input set of univariate time series traces and select its hyperparameters (including the motif length and the edge prediction threshold). Future work may directly investigate uses of MotifDisco on other health datasets.
>
> Personalization (Weakness 2 and Question 4):
> This is an interesting point. The framework does not directly account for inter-individual variability, as it only learns causal relationships across the provided population of input motifs. One way to combat this would be to feed as input only traces from a single individual or from a specific sub-cohort or population, in order to force the framework to focus on understanding personalized causal relationships. This would be quite feasible to do in the diabetes space since glucose traces are typically collected continuously (i.e., every 5 minutes) and there may be months to even years of data per patient.
>
> Recognized Limitations of the Paper (Weakness 3 and 5):
> The reviewer makes valid points about these paper limitations, which we recognize ourselves and address explicitly in Section 7: Conclusion & Limitations. As mentioned in that section, there are many opportunities to further optimize our runtime (e.g., by parallelization). Also, although we do not have a causal ground truth to compare to, our framework performs well on the three use cases, which may provide some evidence that motif causality is useful for real world tasks.
>
> Comparison to SOTA (Weakness 4):
> Please see our response to reviewer t5ma where we have given a longer response to this question. Succinctly, while we do agree that comparing with existing state of the art would make the paper stronger, unfortunately this is not possible because there is no prior work on quantifying causality amongst motifs. We cannot directly compare with prior Granger causal techniques because their framework set-up or model assumptions do not hold for univariate motifs. We have also clarified this in the Related Work section (see lines 122-128).
>
> Clinical Validation (Weakness 6):
> The scope of this paper was focused on describing the theoretical development of the novel motif causality calculation and framework. However, this is a good point and clinical validation is actually the next phase of this project that we are currently working on.

---

> > ### Comment · Reviewer_iup6 · 2024-11-25
> >
> > I appreciate the responses but still think the paper has usability and experimental validation flaws. I won't change my rating.

---

### Official Review · Reviewer_t5ma · 2024-11-04

**Soundness:** 2
**Presentation:** 3
**Contribution:** 2
**Rating:** 5
**Confidence:** 4

**Summary:**

The paper proposes to learn granger causal graphs over time series segments. The segments are defined as motifs as they have specific characteristics. The segments are then embedded into graph node embeddings using GraphSage, a message passing graph neural network. These embeddings are then passed through a link prediction model that maximizes the probability of links with high transfer entropy (conditional entropy gain achieved through adding node i to the conditional to predict node j). This process is then repeated over multiple epochs of the training time series data to learn the graph where links are added iteratively. Subsequently, in each iteration, links are removed by computing the reverse edge graph and their corresponding link prediction probabilities. Finally, this motif causal network is used for time series prediction, anomaly detection and clustering.

**Strengths:**

* Empirically , using the motif causal graphs, improvements in the 3 downstream tasks are identified demonstrate the use of motifs
* Comparison against chunked time series prediction should be presented
* Chunking provides a simple way of making all motifs of the same length

**Weaknesses:**

* Comparing against existing granger causal techniques applied directly on time lagged variables needs to be done to validate the necessity of motifs
* Identifiability of motifs is left as out of scope, but should be discussed as that defines the nodes used in the causal graph construction.
* Metadata such as time of occurrence and frequency of occurrence of motifs is not well presented in an interpretable manner in the link prediction task, how this might be captured in the motif representation is lacking
* A discussion of complex variable length motifs should be presented.

**Questions:**

Comparison with competing Granger causal baselines at various granularity should be presented to improve the soundness of the result.

---

> ### Author Response · Authors · 2024-11-12
> **Response to Reviewer t5ma**
>
> Thanks for your time and effort for this review.
>
> Comparison to SOTA (Question & Weakness 1):
> While we do agree that comparing with existing state of the art would make the paper stronger, unfortunately this is not possible because there is no prior work on quantifying causality amongst motifs. We cannot directly compare with existing Granger causal techniques because their framework set-up or model assumptions do not hold. Specifically, they have one or more of the following issues:
> -	Compose relations between two time series, and cannot find relationships amongst a set of motifs or traces (Amornbunchornvej et al., 2021; Gong et al., 2023).
> -	Formulate causality based on various statistical properties amongst variables- things like variable-based correlation and density-, classification- and prediction-based error measures between variables (Amornbunchornvej et al., 2021; Bonetti et al., 2024; Irribarra et al., 2024). These assumptions do not hold for univariate time series motifs.
> -	Compose causal relationships over time series using repeated statistical measures or temporal dynamics – things like learning implicit repeated patterns, or computing sufficiency or faithfulness hypotheses over lags across the full time series (Sun et al., 2015; Assaad et al., 2021; Löwe et al., 2022; Najafi et al., 2023; Lamp et al., 2024; Pan et al., 2024). These methods do not work for motifs because they are short time series (so there would be no repeated patterns, nor will statistical hypotheses like sufficiency or faithfulness hold since they are not evaluating multiple time lags repeated over a trace.).
> -	Require labels or a known underlying causal structure to guide model training (Gong et al., 2023; Najafi et al., 2023; Bonetti et al., 2024), which is not available for our data and many similar event-based data streams.
>
> To make this more clear, we have updated the Related work section to explicitly state this (see lines 122-128).
>
> Motif Extraction (Weakness 2 and 4):
> The focus of this paper was on the formulation and development of the motif causality framework since that is the novel contribution we bring. As such, discussion about motif extraction methods and artifacts is outside the scope of the paper since the problem of how to extract motifs has been widely solved (e.g., there are many nice methods to pull out motifs in an intelligent manner, given some criteria, and specific to different disciplines, such as Chinpattanakarn & Amornbunchornvej, 2024; Scha ̈fer & Leser, 2022; Ye & Keogh, 2009). Our framework is amenable to any motif extraction method, and, as evidenced by our results, works even when using simple or heuristic motif extraction methods, such as chopping.
>
> Capturing Time Occurrence & Frequency in Framework (Weakness 3):
> Right, so an advantage of Granger Causality is that the model does not need explicit “metadata” about motif temporality to still be able to learn causal relationships amongst data, which is one of the reasons we selected this as our base mathematical notion to build upon.

---

> > ### Comment · Reviewer_t5ma · 2024-11-26
> >
> > Thank you for providing additional context/related work. I would maintain my rating: (1) even without the exact same assumptions being true, the validity of the assumptions made in the paper could be argued by comparing with related work e.g. Tank et al., 2022. (2) Granger causality assumes that time series are stationary - which is a necessary condition to validate - which is why the metadata would be useful.
> >
> > Alex Tank, Ian Covert, Nicholas Foti, Ali Shojaie, and Emily B. Fox. 2022. Neural Granger Causality. IEEE Trans. Pattern Anal. Mach. Intell. 44, 8 (Aug. 2022), 4267–4279. https://doi.org/10.1109/TPAMI.2021.3065601

---

### Meta-Review · Area_Chair_afde · 2024-12-19

**Metareview:**

MotifDisco is a framework that tries to find short segments representing underlying phenomena within time series. MotifDisco uses Granger Causality and Transfer Entropy to learn causal relationships by solving an unsupervised link prediction problem in a graph neural network. I think this is an important problem in healthcare where early identification of signals of future outcomes can be used to create interventions for patients. There were a few problems identified in the manuscript. Reviewers highlighted concerns around identifiability (which was not responded to), the need for evaluations with known ground truth and the limited datasets that the method was evaluated on. I think one method to improve the overall presentation of the work is a pedagogically relevant synthetic data experiment that allows the authors to compare this and other papers on an equal footing that changes the complexity of the underlying dynamics and the hardness of the motif relationship discovery problem. The use of such data would also enable study of the idea in cases where ground truth is known (by construction).

**Additional Comments On Reviewer Discussion:**

The reviewers confirmed that they read the response by the authors but ultimately decided that the work needed further empirical evaluation in order to strengthen its core claims around general applicability.

---

### Decision · Program_Chairs · 2025-01-22

Reject